# AN OPTIMIZATION VIEW ON DYNAMIC ROUTING BETWEEN CAPSULES

**Dilin Wang, Qiang Liu**
Department of Computer Science, University of Texas at Austin
{dilin, lqiang}@cs.utexas.edu

## ABSTRACT

Despite the effectiveness of dynamic routing procedure recently proposed in (Sabour et al., 2017), we still lack a standard formalization of the heuristic and its implications. In this paper, we partially formulate the routing strategy proposed in Sabour et al. (2017) as an optimization problem that minimizes a combination of clustering-like loss and a KL regularization term between the current coupling distribution and its last states. We then introduce another simple routing approach, which enjoys few interesting properties. In an unsupervised perceptual grouping task, we show experimentally that our routing algorithm outperforms the dynamic routing method proposed in Sabour et al. (2017).

## 1 INTRODUCTION

A capsule is a group of neurons (Hinton et al., 2011; Sabour et al., 2017; Hinton et al., 2018), which represents the instantiation parameters of a specific type of an object or an object part. A major advantage of capsules is that they provide a simple way to recognize wholes by recognizing their parts in a way similar to our human perceptual system. In order to get such a part-whole hierarchy off the ground, a dynamic routing mechanism (Sabour et al., 2017) is used to send lower-level (layer $\ell$) capsule predictions to higher-lever (layer $(\ell+1)$) capsules that agrees with the input.

Given a collection of prediction vectors $\{\hat{\mu}_{j|i} = T_{ij}\mu_i\}$ from lower-level capsules, where $\mu_i$ denotes the output of a lower-level capsule $i$ and $T_{ij}$ is a transformation matrix that relates the lower-level capsule $i$ to a higher-level capsule $j$. We denote by $S = \{s_1, \cdots, s_k\}$ the representations of the higher-level capsules, where $s_j$ is in the same feature space as the lower-level capsules predictions $\hat{\mu}_{j|i}$. Let $w_j$ to represent the activation probability of a higher-level capsule $j$ and we assume the weight of each lower-level capsules has been absorbed into $\mu_i$ for simplicity. Let $C = [c_{ij}]_{i,j}$ the coupling probability between capsule $i$ and capsule $j$. Sabour et al. (2017) proposed the following iterative routing procedures to decide how to assign each lower-level capsule predictions to higher-level capsules, as shown in Algorithm 1.

## 2 AN OPTIMIZATION VIEW ON DYNAMIC ROUTING

We observe that the routing procedure proposed in Algorithm 1 could be partially formulated as minimizing a clustering loss function with a KL divergence regularization, defined as follows,

$$\min_{C,S} \left\{ \mathcal{L}(C,S) := -\sum_{i,j} w_j c_{ij} \langle \hat{\mu}_{j|i}, s_j \rangle + \alpha \mathrm{KL}(C||C^{old}) \right\}$$
$$\text{s.t. } c_{ij} > 0, \ \sum_j c_{ij} = 1, \ ||s_j|| \leq 1. \tag{1}$$

where $\langle \cdot, \cdot \rangle$ represents the inner product, and $C^{old} = [c_{ij}^{old}]_{i,j}$ is the coupling probability of the last step. A typical way of solving (1) is to use coordinate descent which optimizes $C$ and $S$ alternatively. Consider the case when $\alpha = 1$, then it is easy to show that update of $S := \{s_j\}$ in Algorithm 1 (Line 4) is equivalent to the coordinate descent on $S$ with $C$ fixed, and the update of $C := \{c_{ij}\}$ (Line 3 & 5) is the coordinate descent on $C$ with $S$ fixed. The caveat of this explanation, however, is that it does not explain the update rule of $w_j := ||\hat{s}_j||^2/(1 + ||\hat{s}_j||^2)$. In the sequel, we propose a new variant of routing procedure which addresses this problem and makes a number of other improvements compared with the original routing algorithm.

---

**Algorithm 1** The Routing Procedure in Sabour et al. (2017)

---

1: for all capsule $i$ in layer $\ell$ and capsule $j$ in layer $(\ell + 1)$: $b_{ij} = 0$
2: **for** iteration $t$ **do**
3:      for all capsule $i$ in layer $\ell$: $c_{ij} = \frac{\exp(b_{ij})}{\sum_k \exp(b_{ik})}$
4:      for all capsule $j$ in layer $(\ell + 1)$: $\hat{s}_j = \sum_i c_{ij}\hat{\mu}_{j|i}, \quad s_j = \hat{s}_j/||\hat{s}_j||$.
5:      for all capsule $i$ in layer $\ell$ and $j$ in layer $(\ell+1)$: $b_{ij} = b_{ij} + w_j\langle\hat{\mu}_{j|i}, s_j\rangle$, where $w_j = \frac{||\hat{s}_j||^2}{1+||\hat{s}_j||^2}$.
6: **end for**
7: **Return** $w_j s_j$

---

**Algorithm 2** Our Routing Algorithm

---

1: **for** iteration $t$ **do**
2:      for all capsule $i$ in layer $\ell$ and capsule $j$ in layer $(\ell + 1)$: $b_{ij} = \frac{1}{\alpha}\langle o_{j|i}, s_j\rangle \quad c_{ij} = \frac{\exp(b_{ij})}{\sum_k \exp(b_{ik})}$.
3:      for all capsule $j$ in layer $(\ell + 1)$: $\hat{s}_j = \sum_i c_{ij}o_{j|i}, \quad s_j = \hat{s}_j/||\hat{s}_j||$.
4: **end for**
5: for all capsule $j$ in layer $(\ell + 1)$: $w_j = \frac{||\sum_i c_{ij}o_{j|i}||}{1+\max_k||\sum_i c_{ik}o_{k|i}||}$
6: **Return** $w_j s_j$

---

## 3    OUR APPROACH

Our algorithm is summarized in Algorithm 2. It is motivated as solving the following clustering-like objective function:

$$\min_{C,S} \left\{ \mathcal{L}(C,S) := -\sum_i \sum_j c_{ij}\langle o_{j|i}, s_j\rangle + \alpha \sum_i \sum_j c_{ij}\log c_{ij} \right\}, \tag{2}$$
$$\text{s.t. } \sum_j c_{ij} = 1, c_{ij} > 0, ||s_j|| \leq 1$$

where $o_{j|i} = \frac{1}{||T_{ij}||_{\mathcal{F}}}T_{ij}\mu_i$ and $||T_{ij}||_{\mathcal{F}}$ represents the Frobenius norm of $T_{ij}$. Our objective is similar to agglomerative fuzzy K-Means algorithm (Li et al., 2008). Deriving the coordinate descent updates of $C$ and $S$, we obtain the updates in Algorithm 2.

**Update of $w_j$**    Compared with (1), we remove the dependency of activation probability $w_j$ from the objective (2), and set the $w_j$ at the end of the routing procedure instead. In this way, we formulate our routing algorithm as a more formal optimization problem. In addition, setting $w_j$ only at the end of the procedure may prevent $w$ to become highly unbalanced as the iteration number increases.

**Scale-invariant**    Another modification we made is to normalize the transformation matrix $T_{ij}$ before inputting it into the procedure, that is, we use $o_{j|i} := \frac{1}{||T_{ij}||_{\mathcal{F}}}T_{ij}\mu_i$ as the input. Note $\hat{\mu}_{j|i} = T_{ij}\mu_i$, assume $||\mu_i|| \leq 1$, in order to stabilize the whole training process, one need to regularize on transformation matrix $T_{ij}$, This prevents the LHS of Eq. (1) goes to infinite negative by as we learn the values of $T_{ij}$ during trained. The objective in (1) does not prevent the problem by itself, instead, Sabour et al. (2017) addressed this problem by using an interesting margin loss. Specifically, if an entity is present by capsule $j$, then a loss of $\max(0, m^+ - w_j)^2$ is applied. Otherwise, the loss is taken to be $\max(0, w_j - m^-)^2$, where $m^+ = 0.9, m^- = 0.1$ respectively. This is equivalent to force

$$\frac{m^-}{1 - m^-} \leq ||\sum_i c_{ij}T_{ij}\mu_i||^2 \leq \frac{m^+}{1 - m^+}. \tag{3}$$

In this paper, we propose a more general way to do regularization. For each routing iteration, we re-scale the weight matrix $T_{ij}$, and the scale of inner product between $\langle\frac{1}{||T_{ij}||_{\mathcal{F}}}T_{ij}\mu_i, s_j\rangle$ is restricted to a value below 1. And the activation weight $w_j$ doesn't depend on the scale of $T_{ij}$ either.

**Annealing the Regularization Term**  Note that Sabour et al. (2017) used a $KL(C||C^{old})$ regularization with a fixed coefficient $\alpha = 1$. We propose to replace the KL divergence with an entropy regularization, which enforces $c_{ij}$ to close to uniform, instead of its previous value; this makes the output of routing depends on the input in a more smooth way and hence stablizes the algorithm. We also gradually decrease the value of $\alpha$ during the iterations. Intuitively, In the early phase, the predictions of lower-level capsules are not reliable and we need to update the network parameters to obtain more discriminative representations for the subsequent routing processes. Therefore, we should set a large $\alpha$ and to make the entropy play a more important role, so that the routing process will try to assign each lower-level capsules to more higher-level capsules in a more uniform way; In the late stage of training, we should set a small $\alpha$, so that the routing process will try to maximize the agreements between lower-level capsules and higher-level capsules, so that $c_{ij}$ need to be more deterministic. This is similar to assign each lower-level capsule to its nearest higher-level capsule. We find that the value of $\alpha$ is crucial for the performance in lots of unsupervised tasks, we leave a thorough discussion to our future work.

## 4    EXPERIMENTAL RESULTS

In section, we evaluate the performance of our agglomerative routing approach on a simple unsupervised perceptual grouping task that involves grouping three randomly chosen regular shapes ($\triangle, \triangledown, \square$) located in random positions of $28 \times 28$ images. Following the settings in Greff et al. (2017), Each input image can be thought as a spatial mixture of $k$ components parametric by representations $\{s_1, \cdots, s_k\}$. A neural network $f_\theta$ is trained to transform these capsule $\{s_j\}$ into pixel-wise predictions. The desired goal is to train a Capsule-network that produces coherent explanations, which could be further used to decode each object in the inputs respectively.

In our experiments, we use the same Capsule-network structure as the one used in (Sabour et al., 2017). The representations $\{s_k\}$ is real-valued 16 dimensional vectors generated by the capsule networks, we fix $k = 4$. Let $m_k$ be the group assignment probabilities and $z_k = f_\theta(s_k)$ be the expected prediction pixel value for that group. Both $m_k$ and $z_k$ has the same dimension as the input image $x$. In this way, the final reconstruction loss is defined as the difference between the original inputs $x$ and the averaged summation over all predictions

$$\mathcal{L} = \sqrt{\sum_i (x_i - \sum_k m_{k,i} z_{k,i})^2}, \quad \text{s.t. } m_{k,i} \geq 0, \quad \sum_k m_{k,i} = 1.$$

We evaluate the quality of the learned groupings with respect to the ground truth while ignoring the background and overlap regions in a way similar to Greff et al. (2017). We can see from table 4 that our routing approach achieves better performance than other baselines.

|  | Routing =3 | Routing = 5 | Routing =10 | Routing =15 |
|---|---|---|---|---|
| (Sabour et al., 2017) | 0.647 | 0.830 | 0.862 | 0.879 |
| Our Approach | 0.816 | 0.901 | 0.911 | 0.914 |
| N-EM (Greff et al., 2017)* | | 0.475 | | |
| RNN-EM (Greff et al., 2017)* | | 0.826 | | |

Table 1:  Adjusted Mutual Information (AMI) score ($[0, 1]$) (Vinh et al., 2010), the higher the better. * as reported in (Greff et al., 2017).

## 5    CONCLUSION

We show that the routing mechanism proposed in (Sabour et al., 2017) is similar to minimize a standard clustering loss with a KL regularization on the coupling probabilities. We discuss few possible ways to improve the performance of capsule networks. Future work includes clear discussions and experiments on larger datasets.

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
