# OpenReview forum: "An Optimization View on Dynamic Routing Between Capsules"
_ICLR.cc/2018/Workshop — Accept_

### Official Review · AnonReviewer2 · 2018-03-07
**can you test this on the same benchmarks as the capsules paper?**

**Rating:** 5
**Confidence:** 3

**Review:**

The paper provides an interpretation of the routing method, which is interesting. However, having dependence on C_old makes it not exactly a loss function.

The presented result seems convincing. However, the author ought to compare using tasks on the capsules paper, which Sabour et al used to develop their routing method. If there is compelling reasons not to do this,  the authors should state those reasons.

---

### Official Review · AnonReviewer1 · 2018-03-12
**Interesting analysis of recently proposed algorithm**

**Rating:** 7
**Confidence:** 3

**Review:**

The paper analyzes a recently proposed algorithm in Dynamic Routing between Capsules (Sabour et al. 2017).
The original dynamic routing algorithm proposes a heuristic to determine the connectivity weights of capsules in a lower level to those in a higher level. This paper proposes a modified version of this algorithm and can be motivated by minimizing a loss. I think the paper is an interesting analysis/follow up work and should be accepted as a workshop paper.

---

### Public Comment · ~Hongyang_Li1 · 2018-02-28
**Clarification**

Hi, could you explain a little bit more why the nips paper is trying to minimize the problem as Eqn. (1)? Specifically, the output capsule is w_j*s_j in your formulation where w_j is part of the squash operation. But how does it end up with minimizing -\sum w*c*<\hat_u, s>? what does it mean? If I understand correctly, w*<\hat_u, s> is the \delta_b (update rule, line 5 in Algorithm 1), so the optimization becomes -\sum c * \delta_b ?

Also do you try replacing the proposed routing on CIFAR or MNIST?

---

> ### Public Comment · ~Dilin_Wang2 · 2018-03-09
> **optimization**
>
> Given C and w_j fixed, minizing eq (1) w.r.t. s_j is equivelent to \min -\sum_i <c_ij \hat u_ji, s_j>,
> which by Cauchy-Schwartz is minimized  iff s_j \propto \sum_i <c_ij \hat u_ji>; this is the update rule as in line 4 (algrothm 1); given s and w_j fixed,  c_{ij} \propto c^{old}_{ij}\exp(w_j <\hat u_ji, s_j>), replace c_{ij} = exp(b_ij), we have
> b_ij = b_ij^old + w_j <\hat u_ji, s_j>, c_{ij} = exp(b_ij)/ \sum_j exp(b_ij)
>
> We're working on more benchmarks. Thanks.

---

### Decision · Program_Chairs · 2018-03-20
**ICLR 2018 Workshop Acceptance Decision**

**Decision:**

Accept

**Comment:**

Congratulations, your paper was accepted to the ICLR workshop.